# LIGHTWEIGHT CONVOLUTIONAL NEURAL NETWORKS BY HYPERCOMPLEX PARAMETERIZATION

## ABSTRACT

Hypercomplex neural networks have proved to reduce the overall number of parameters while ensuring valuable performances by leveraging the properties of Clifford algebras. Recently, hypercomplex linear layers have been further improved by involving efficient parameterized Kronecker products. In this paper, we define the parameterization of hypercomplex convolutional layers to develop lightweight and efficient large-scale convolutional models. Our method grasps the convolution rules and the filters organization directly from data without requiring a rigidly predefined domain structure to follow. The proposed approach is flexible to operate in any user-defined or tuned domain, from 1D to $n$D regardless of whether the algebra rules are preset. Such a malleability allows processing multidimensional inputs in their natural domain without annexing further dimensions, as done, instead, in quaternion neural networks for 3D inputs like color images. As a result, the proposed method operates with $1/n$ free parameters as regards its analog in the real domain. We demonstrate the versatility of this approach to multiple domains of application by performing experiments on various image datasets as well as audio datasets in which our method outperforms real and quaternion-valued counterparts.

## 1 INTRODUCTION

Recent state-of-the-art convolutional models achieved astonishing results in various fields of application by large-scaling the overall parameters amount (Karras et al., 2020; d'Ascoli et al., 2021; Dosovitskiy et al., 2021). Simultaneously, quaternion neural networks (QNNs) demonstrated to significantly reduce the number of parameters while still gaining comparable performances (Parcollet et al., 2019c; Grassucci et al., 2021a; Tay et al., 2019). Quaternion models exploit hypercomplex algebra properties, including the Hamilton product, to painstakingly design interactions among the imaginary units, thus involving $1/4$ of free parameters with respect to real-valued models. Furthermore, thanks to the modelled interactions, quaternion networks capture internal latent relations in multidimensional inputs and preserve pre-existing correlations among dimensions. Therefore, the quaternion domain is particularly appropriate for processing 3D or 4D data, such as color images or (up to) 4-channel signals. Unfortunately, most common color image datasets contain RGB images and a zero channel has to be padded to the input in order to encapsulate the image in the four quaternion components. Additionally, while quaternion neural components are widespread and easy to be integrated in pre-existing models, very few attempts have been made to extend models to higher domain orders. Accordingly, the development of hypercomplex convolutional models for larger multidimensional inputs, such as magnitudes and phases of multichannel audio signals or 16-band satellite images, still remains painful. Recently, a parameterized hypercomplex multiplication (PHM) for fully connected layers have been proposed to generalize hypercomplex multiplications as sum of Kronecker products, going beyond quaternion algebra. However, no solution exists for convolutional layers, which remain the most employed layers when dealing with multidimensional inputs, such as images and audio signals (Wu et al., 2021; Hershey et al., 2017).

In this paper, we propose a parameterized hypercomplex convolutional (PHC) layer to define lightweight large-scale neural models admitting any multidimensional input, whichever the number of dimensions. Our method is flexible to operate in domains from 1D to $n$D, where $n$ can be arbitrarily chosen by the user or tuned to let the model performance lead to the most appropriate domain for the given input data. Such a malleability comes from the ability of the proposed approach

to subsume algebra rules to perform convolution regardless of whether these regulations are preset or not. Thus, neural models endowed with our approach adopt $1/n$ of free parameters with respect to their real-valued counterparts, and the amount of parameter reduction is a user choice. This makes PHC layers adaptable to a plethora of applications and hardware, even on the edge where memory saving is a crucial aspect. Additionally, the PHC versatility allows processing multidimensional data in its natural domain by simply setting the dimensional hyperparameter $n$. For instance, color images can be analyzed in their RGB domain by setting $n = 3$ without adding any useless information, contrary to standard processing for quaternion networks with the padded zero-channel. Indeed, PHC-based models are able to grasp the proper algebra from input data, while capturing internal correlations among the image channels and saving $66\%$ of free parameters.

On a thorough empirical evaluation on multiple benchmarks, we demonstrate the flexibility of our method that can be adopted in different domain of applications, from images to audio signals. We devise a set of convolutional models endowed with PHC layers for large-scale image classification and sound event detection tasks, letting them operate in different hypercomplex domain and with various input dimensionality with $n$ ranging from 2 to 16.

The contribution of this paper is three-fold.

- We introduce a parameterized hypercomplex convolutional (PHC) layer, which aims at building lightweight and more efficient large-scale convolutional models. The approach grasps the convolution rules directly from data via backpropagation exploiting the Kronecker product properties, thus reducing the number of free parameters to $1/n$.

- We show how the proposed approach can be employed with any kind of multidimensional data by easily changing the hyperparameter $n$. Indeed, by setting $n = 3$ the PHC-based models can process RGB images in their natural domain, while leveraging the properties of hypercomplex algebras, allowing parameters sharing inside the layers and leading to a parameter reduction to $1/3$. To the best of our knowledge, this is the first approach that processes color images with hypercomplex-based neural models without adding any padding channel. As well, multichannel audio signals can be analysed by simply considering $n = 4$ for standard first-order ambisonics (which has 4 microphone capsules), $n = 8$ for an array of two ambisonics microphones, or even $n = 16$ if we want to include the information of each channel phase.

- We devise a family of PHC neural networks, redefining common ResNets, VGGs and Sound Event Detection networks (SEDnets), operating in any user-defined domain just by choosing the hyperparameter $n$, which also drives the number of convolutional filters. [1]

The rest of the paper is organized as follows. In Section 2, we recapitulate real and quaternion-valued convolutional layers. Section 3 theoretically introduces the proposed method and expounds how to process RGB images with $n = 3$, while in Section 4 we define the PHC models and we present the experimental evaluation, including datasets, experiment details and results. Finally, Section 5 reports the related works and in Section 6 we draw conclusions.

## 2 Real and Quaternion-Valued Convolutional Layers

A brief recall on real-valued and quaternion convolutional layers is appropriate to better understand the proposed method. A generic convolutional layer can be described by

$$\mathbf{y} = \mathrm{Conv}(\mathbf{x}) = \mathbf{W} * \mathbf{x} + \mathbf{b}, \tag{1}$$

where the input $\mathbf{x} \in \mathbb{R}^{t \times s}$ is convolved ($*$) with the filter tensor $\mathbf{W} \in \mathbb{R}^{s \times d \times k \times k}$ to produce the output $\mathbf{y} \in \mathbb{R}^{d \times t}$, where $k$ is the filter size. The bias term $\mathbf{b}$ does not heavily influence the number of parameters, thus the degrees of freedom for this operation are essentially $\mathcal{O}(sdk^2)$.

Quaternion convolutional layers, instead, build the weight tensor $\mathbf{W} \in \mathbb{R}^{s \times d \times k \times k}$ by following the Hamilton product rule and organize filters according to it:

---

[1]Full code is available at: `https://anonymous.4open.science/r/HyperNets-CBBB`.

$$\mathbf{W} * \mathbf{x} = \begin{bmatrix} \mathbf{W}_0 - \mathbf{W}_1 - \mathbf{W}_2 - \mathbf{W}_3 \\ \mathbf{W}_1 + \mathbf{W}_0 - \mathbf{W}_3 + \mathbf{W}_2 \\ \mathbf{W}_2 + \mathbf{W}_3 + \mathbf{W}_0 - \mathbf{W}_1 \\ \mathbf{W}_3 - \mathbf{W}_2 + \mathbf{W}_1 + \mathbf{W}_0 \end{bmatrix} * \begin{bmatrix} \mathbf{x}_0 \\ \mathbf{x}_1 \\ \mathbf{x}_2 \\ \mathbf{x}_3 \end{bmatrix}, \tag{2}$$

where $\mathbf{W}_0, \mathbf{W}_1, \mathbf{W}_2, \mathbf{W}_3 \in \mathbb{R}^{\frac{s}{4} \times \frac{d}{4} \times k \times k}$ are the real coefficients of the quaternion weight matrix $\mathbf{W} = \mathbf{W}_0 + \mathbf{W}_1\hat{\imath} + \mathbf{W}_2\hat{\jmath} + \mathbf{W}_3\hat{\kappa}$ and $\mathbf{x}_0, \mathbf{x}_1, \mathbf{x}_2, \mathbf{x}_3$ are the coefficients of the quaternion input $\mathbf{x}$ with the same structure. The imaginary units comply with the property $\hat{\imath}^2 = \hat{\jmath}^2 = \hat{\kappa}^2 = -1$ and with the non-commutative products $\hat{\imath}\hat{\jmath} = -\hat{\jmath}\hat{\imath}; \; \hat{\jmath}\hat{\kappa} = -\hat{\kappa}\hat{\jmath}; \; \hat{\kappa}\hat{\imath} = -\hat{\imath}\hat{\kappa}$.

As done for real-valued layers, the bias can be ignored and the degree of freedom computations of the quaternion convolutional layer can be approximated to $\mathcal{O}(sdk^2/4)$. The lower number of parameters with respect to the real-valued operation is due to the reuse of filters performed by the Hamilton product in Eq.2. Also, sharing the parameter submatrices forces to consider and exploit the correlation between the input components (Parcollet et al., 2019a; Tay et al., 2019; Gaudet & Maida, 2018).

## 3 PARAMETERIZING HYPERCOMPLEX CONVOLUTIONS

### 3.1 PARAMETERIZED HYPERCOMPLEX CONVOLUTIONAL LAYERS

In the following, we delineate the formulation for the proposed parameterized hypercomplex convolutional (PHC) layer. We also show that this approach is capable of learning the Hamilton product rule when two quaternions are convolved. The PHC layer is based on the construction, by sum of Kronecker products, of the weight tensor $\mathbf{H}$ which encapsulates and organizes the filters of the convolution. The proposed method is formally defined as:

$$\mathbf{y} = \mathrm{PHC}(\mathbf{x}) = \mathbf{H} * \mathbf{x} + \mathbf{b}, \tag{3}$$

whereby $\mathbf{H} \in \mathbb{R}^{s \times d \times k \times k}$ is built by sum of Kronecker products between two learnable matrices. Here, $s$ is the input dimensionality to the layer, $d$ is the output one, and $k$ is the filter size. More concretely,

$$\mathbf{H} = \sum_{i=1}^{n} \mathbf{A}_i \otimes \mathbf{F}_i, \tag{4}$$

in which $\mathbf{A}_i \in \mathbb{R}^{n \times n}$ with $i = 1, ..., n$ are the matrices that describe the algebra rules and $\mathbf{F}_i \in \mathbb{R}^{\frac{s}{n} \times \frac{d}{n} \times k \times k}$ represents the $i$-th batch of filters that are arranged by following the algebra rules to compose the final weight matrix. The core element of this approach is the Kronecker product, which is a generalization of the vector outer product that can be parameterized by $n$. The hyperparameter $n$ can be set by the user who wants to operate in a pre-defined real or hypercomplex domain (e.g., by setting $n = 2$ the PHC layer is defined in the complex domain, or in the quaternion one if $n$

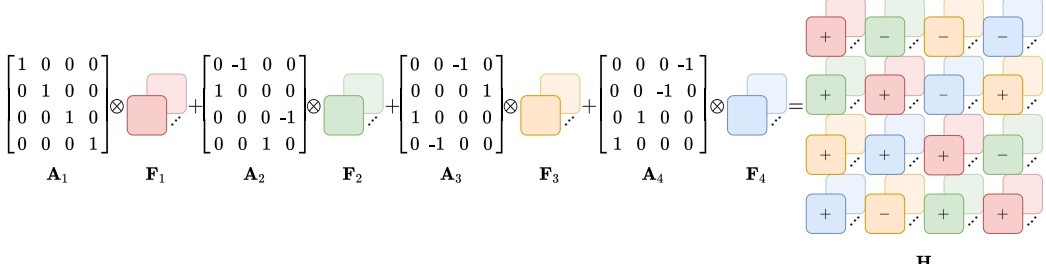

Figure 1: The quaternion convolution rule can be expressed as sum of Kronecker products between the matrices $\mathbf{A}_i$ that subsume the algebra rules and the matrices $\mathbf{F}_i$ that contain the convolution filters, with $i = 1, 2, 3, 4$. In this example, the parameters of $\mathbf{A}_i$ are fixed for visualization purposes, but in PHC layers they are learnable parameters.

is set equal to $4$, as Figure 1 illustrates), or tuned to obtain the best performance from the model. The matrices $\mathbf{A}_i$ and $\mathbf{F}_i$ are learnt during training and their values are reused to build the definitive tensor $\mathbf{H}$.

The degree of freedom of $\mathbf{A}_i$ and $\mathbf{F}_i$ are $n^3$ and $sdk^2/n$, respectively. Usually, real world applications employ a large number of filters in layers ($s, d = 256, 512, ...$) and small values for $k$. Therefore, frequently $sdk^2 \gg n^3$ holds. Thus, the degrees of freedom for the PHC weight matrix can be approximated to $\mathcal{O}(sdk^2/n)$. Hence, the PHC layer reduces the number of parameters by $1/n$ with respect to a standard convolutional layer in real world problems.

Moreover, when processing multidimensional data with correlated channels, such as color images, rather than mulichannel audio or multisensor signals, PHC layers bring benefits due to the weight sharing among different channels. This allows capturing latent intra-channels relations that standard convolutional networks ignore because of the rigid structure of the weights (Grassucci et al., 2021a; Parcollet et al., 2019b). The PHC layer is able to subsume hypercomplex convolution rules and the desired domain is specified by the hyperparameter $n$. We test this ability with toy problems in Appendix B. Interestingly, by setting $n = 1$ a real-valued convolutional layer can be represented too. Indeed, standard real layers does not involve parameters sharing, therefore the algebra rules are solely described by the single $\mathbf{A} \in \mathbb{R}^{1 \times 1}$ and the complete set of filters are included in $\mathbf{F}^{s \times d \times k \times k}$. We report detailed explanations in Appendix A.

## 3.2 PARAMETERIZED LAYERS FOR COLOR IMAGES

In this section, we describe how the PHC layer can be employed to process color images in hypercomplex domains without needing any additional information to the input. Different encodes exist to process color images, however, the most common computer vision datasets are comprised of three-channel images in $\mathbb{R}^3$. In the quaternion domain, RGB images are enclosed into a quaternion and processed as single elements (Parcollet et al., 2019a). The encapsulation is performed by considering the RGB channels as the real coefficients of the imaginary units and by padding a zeros channel as the first real component of the quaternion.

Here, we propose to leverage the high malleability of PHC layers to deal with RGB images in hypercomplex domains without embedding useless information to the input. Indeed, the PHC can directly operate in $\mathbb{R}^3$ by easily setting $n = 3$ and process RGB images in their natural domain while exploiting hypercomplex algebra properties such as parameters sharing.

Indeed, the great flexibility of PHC layers allows the user to choose whether processing images in $\mathbb{R}^4$ or $\mathbb{R}^3$. On one hand, by setting $n = 4$, the zeros channel is added to the input even so the layer saves the $75\%$ of free parameters. On the other hand, by choosing $n = 3$ the network does not handle any useless information, notwithstanding, it reduces the number of parameters by solely $66\%$. This is a trade-off which may depend on the application or on the hardware the user needs. Furthermore, the domain on which processing images can be tuned by letting the performance of the network indicates the best choice for $n$.

## 3.3 PARAMETERIZED LAYERS FOR MULTICHANNEL AUDIO

In the following, we expound how the proposed PHC layer can be employed to deal with multichannel audio signals. For instance, first-order Ambisonics (FOA) is composed of $4$ microphone capsules, whose magnitude representations can be enclosed in a quaternion (Comminiello et al., 2019; Ricciardi Celsi et al., 2020). However, the quaternion algebra may be restrictive if more than one microphone is employed for registration or whether the phase information has to be included too. Indeed, quaternion neural networks badly fit with multidimensional input with more than $4$ channels.

Conversely, the proposed method can be easily adapted to deal with these additional dimensions by handily setting the hyperparameter $n$ and thus completely leveraging each information in the $n$-dimensional input.

# 4 CONVOLUTION-BASED NEURAL MODELS WITH PHC LAYERS

In this section, we expound neural models endowed with PHC layers and we demonstrate how the proposed method can be employed in various spheres of application. To this end, we devise deep PHC models for image classification and for sound event detection. For each task, we empirically substantiate the effectiveness and flexibility of the proposed PHC layer. In order to be consistent with the literature, we perform each experiment with a real-valued baseline model and then we compare it with its quaternion counterpart and with the proposed PHC network. Furthermore, we assess the PHC malleability testing different values of the hyperparameter $n$, therefore defining PHC models in multiple hypercomplex domains.

## 4.1 HYPERCOMPLEX PARAMETERIZATION FOR IMAGE CLASSIFICATION

To begin with, we test the PHC layer on RGB images and we show how, exploiting the correlations among channels, the proposed method saves parameters while ensuring high performances.

### 4.1.1 PARAMETERIZED HYPERCOMPLEX RESNETS

In recent literature, a copious set of high performance in image classification is obtained with models having a residual structure. ResNets (He et al., 2016) pile up manifold residual blocks composed of convolutional layers and identity mappings. A generic PHC residual block is defined by

$$\mathbf{y} = \mathcal{F}(\mathbf{x}, \{\mathbf{H}_j\}) + \mathbf{x}, \tag{5}$$

whereby $\mathbf{H}_j$ are the PHC weights of layer $j = 1, 2$ in the block, and $\mathcal{F}$ is

$$\mathcal{F}(\mathbf{x}, \{\mathbf{H}_j\}) = \text{PHC}\left(\text{ReLU}\left(\text{PHC}(\mathbf{x})\right)\right). \tag{6}$$

### 4.1.2 PARAMETERIZED HYPERCOMPLEX VGGS

Another family of popular methods for image classification is based on the VGG networks (Simonyan & Zisserman, 2015) that stack several convolutional layers and a closing fully connected classifier. To completely define models in the desired hypercomplex domain, we propose to endow the network with PHC layers as convolution components and with Parameterized Hypercomplex Multiplication (PHM) layers (Zhang et al., 2021) as linear classifier. The backbone of our PHC VGG is then

$$\begin{aligned}
\mathbf{h}_t &= \text{ReLU}\left(\text{PHC}_t\left(\mathbf{h}_{t-1}\right)\right) \qquad t = 1, ..., j \\
\mathbf{y} &= \text{ReLU}\left(\text{PHM}(\mathbf{h}_j)\right).
\end{aligned} \tag{7}$$

We also test a hybrid model by employing standard real-valued fully connected (FC) layers instead of PHM ones. However, we believe that it is out of the scope of this paper which aims at exploiting hypercomplex algebras properties to define highly malleable neural networks. Nevertheless, for a further comparison, we report these additional experiments in the Appendix B.

### 4.1.3 EXPERIMENTAL SETUP AND EVALUATION

We perform the image classification task with five baseline models. We consider ResNet18, ResNet50 and ResNet152 from the ResNet family and VGG16 and VGG19 from the VGG one. Each hyperparameter is set according to the original papers (He et al., 2016; Simonyan & Zisserman, 2015). We investigate the performance in four different color images datasets at different scales. We employ SVHN, CIFAR10, CIFAR100, and ImageNet and any kind of data augmentation is applied to these datasets in order to guarantee a fair comparison.

We execute initial experiments with VGGs against Quaternion VGGs and two versions of PHC VGGs with $n$ equal to 2 and to 4. Average and standard deviation accuracy over three runs are reported for SVHN and CIFAR10 datasets in Table 1. Both the VGG16 and VGG19 PHC-based versions clearly outperform real and quaternion counterparts while being built with more than a half the number of parameters of the baseline. Additionally, PHC-based models extraordinarily reduce the number of training and inference time required with respect to the quaternion model which operates in a hypercomplex domain as well. Furthermore, when scaling up the experiment with VGG19,

Table 1: Image classification results for VGG. The accuracy mean and standard deviation over three runs with different seeds is reported. Training (T) time and inference (I) time required on CIFAR10. For training time we report, in seconds per 100 iterations, the mean and the standard deviation over the iterations in one epoch, while the inference time is the time required to decode the test set. The PHC model with $n = 4$ outperforms the quaternion counterpart both in terms of accuracy and time. The PHC with $n = 2$ far exceeds the real-valued baseline in the considered datasets, while both the PHC VGG19 versions with $n = 2, 4$ are more efficient than the real and quaternion-valued baselines at inference time.

| Model | Params | SVHN | CIFAR10 | Time (T) | Time (I) |
|---|---|---|---|---|---|
| VGG16 | 15M | $94.364 \pm 0.394$ | $85.067 \pm 0.765$ | $\mathbf{2.2 \pm 0.02}$ | **1.2** |
| Quaternion VGG16 | 3.8M (-75%) | $93.887 \pm 0.292$ | $83.997 \pm 0.493$ | $5.2 \pm 0.02$ | 2.2 |
| PHC VGG16 $n = 2$ | 7.6M (-50%) | $\mathbf{94.831 \pm 0.257}$ | $\mathbf{86.510 \pm 0.216}$ | $3.2 \pm 0.02$ | 1.4 |
| PHC VGG16 $n = 4$ | 3.8M (-75%) | $94.639 \pm 0.121$ | $85.640 \pm 0.205$ | $3.2 \pm 0.02$ | 1.4 |
| VGG19 | 29.8M | $94.140 \pm 0.129$ | $85.624 \pm 0.257$ | $\mathbf{3.2 \pm 0.02}$ | 16.0 |
| Quaternion VGG19 | 7.5M (-75%) | $93.983 \pm 0.190$ | $83.914 \pm 0.129$ | $6.2 \pm 0.02$ | 16.3 |
| PHC VGG19 $n = 2$ | 14.9M (-50%) | $\mathbf{94.553 \pm 0.229}$ | $\mathbf{85.750 \pm 0.286}$ | $4.0 \pm 0.02$ | **15.4** |
| PHC VGG19 $n = 4$ | 7.4M (-75%) | $94.169 \pm 0.296$ | $84.830 \pm 0.733$ | $4.2 \pm 0.02$ | 15.5 |

Table 2: Image Classification results with ResNet models. Each experiment is run three times with different seeds and mean with standard deviation is reported. The proposed models far exceed real-valued and quaternion baselines almost in each experiment we conduct. Interestingly, the PHC model outperform the real-valued counterpart by $4\%$ points in the largest-scale experiment on CIFAR100. The time is similar to the claims in Table 1 so we do not add here to avoid redundancy.

| Model | Params | FLOPs | SVHN | CIFAR10 | CIFAR100 |
|---|---|---|---|---|---|
| ResNet18 | 10.1M | 1.01G | $93.992 \pm 1.317$ | $89.543 \pm 0.340$ | $62.634 \pm 0.600$ |
| Quaternion ResNet18 | 2.8M (-75%) | 1.01G | $93.661 \pm 0.413$ | $88.240 \pm 0.377$ | $59.850 \pm 0.607$ |
| PHC ResNet18 $n = 2$ | 5.4M (-50%) | 1.03G | $\mathbf{94.359 \pm 0.187}$ | $89.260 \pm 0.625$ | $60.320 \pm 2.249$ |
| PHC ResNet18 $n = 3$ | 3.6M (-66%) | 1.03G | $94.303 \pm 1.234$ | $\mathbf{89.603 \pm 0.563}$ | $\mathbf{62.660 \pm 1.067}$ |
| PHC ResNet18 $n = 4$ | 2.7M (-75%) | 1.03G | $94.234 \pm 0.161$ | $88.847 \pm 0.874$ | $61.780 \pm 0.689$ |
| ResNet50 | 22.5M | 2.36G | $94.546 \pm 0.269$ | $89.630 \pm 0.305$ | $65.514 \pm 0.569$ |
| Quaternion ResNet50 | 5.7M (-75%) | 2.36G | $93.685 \pm 0.389$ | $89.670 \pm 0.383$ | $63.760 \pm 0.717$ |
| PHC ResNet50 $n = 2$ | 11.1M (-50%) | 2.41G | $93.849 \pm 0.249$ | $89.750 \pm 0.386$ | $65.884 \pm 0.333$ |
| PHC ResNet50 $n = 3$ | 7.6M (-66%) | 2.41G | $93.617 \pm 0.497$ | $\mathbf{90.423 \pm 0.145}$ | $\mathbf{66.497 \pm 1.256}$ |
| PHC ResNet50 $n = 4$ | 5.7M (-75%) | 2.41G | $\mathbf{94.558 \pm 0.754}$ | $88.897 \pm 0.645$ | $66.240 \pm 1.165$ |
| ResNet152 | 52.6M | 6.62G | $\mathbf{94.625 \pm 0.355}$ | $89.580 \pm 0.173$ | $62.053 \pm 0.385$ |
| Quaternion ResNet152 | 13.2M (-75%) | 6.62G | $93.638 \pm 0.098$ | $89.227 \pm 0.287$ | $61.267 \pm 0.784$ |
| PHC ResNet152 $n = 2$ | 26.6M (-50%) | 6.76G | $93.915 \pm 0.512$ | $\mathbf{90.540 \pm 0.401}$ | $65.817 \pm 0.327$ |
| PHC ResNet152 $n = 3$ | 17.8M (-66%) | 6.76G | $93.955 \pm 0.152$ | $90.077 \pm 0.436$ | $66.347 \pm 0.567$ |
| PHC ResNet152 $n = 4$ | 13.4M (-75%) | 6.76G | $94.290 \pm 0.237$ | $89.897 \pm 0.097$ | $\mathbf{66.437 \pm 0.064}$ |

the proposed methods are more efficient at inference time with respect to the real-valued VGG19. Therefore, PHC models can be easily adopted in applications with disk memory limitations, due to the reduction of parameters, and for fast inference problems thanks to the efficiency at testing time. More initial experiments with various networks on other datasets are reported in Appendix B.

The malleability of the proposed approach when dealing with color images is expressed in the opportunity to choose the domain in which operating. Therefore, we test PHC networks in the complex $\mathbb{H}^2$ ($n = 2$), quaternion $\mathbb{H}^4$ ($n = 4$) or $\mathbb{H}^3$ ($n = 3$) domain, where in the latter we do not concatenate any zero padding and process the RGB channels of the image in their natural domain. In order to implement this experiment, we change the number of filters in the baseline and in the corresponding compared networks to be divisible by each value of $n$. More details about the implementations are provided in Appendix B and in the GitHub repository (link in Section 1).

Table 3: Storage memory required for ResNet152s checkpoints on CIFAR100. Quaternion and PHC models allow a considerable disk memory saving with respect to the real-valued ResNet.

| Model | Storage Memory |
|---|---|
| ResNet152 | 201 MB |
| Quaternion ResNet152 | 51 MB (-75%) |
| PHC ResNet152 $n = 2$ | 103 MB (-49%) |
| PHC ResNet152 $n = 3$ | 70 MB (-65%) |
| PHC ResNet152 $n = 4$ | 53 MB (-74%) |

Table 4: ImageNet classification with real-valued baseline against our best model PHC $n = 3$. Our approach outperform the baseline while saving the $66\%$ of parameters.

| Model | Params | ImageNet |
|---|---|---|
| ResNet50 | 25.7M | 67.990 |
| PHC ResNet50 $n = 3$ | 9.6M (-66%) | **68.584** |

Table 2 presents the the mean and standard deviation accuracy over three runs with different seeds for the ResNet-based models. We perform extensive experiments and the PHC models with $n = 4$ always outperform the quaternion counterpart gaining a higher accuracy and being more robust. This underlines the effectiveness of the PHC architectural flexibility over the predefined and rigid structure of quaternion layers. Furthermore, our method distinctly far exceeds the corresponding real-valued baselines across the experiments while saving from $50\%$ to $75\%$ parameters. Focusing on the latter result, the PHC model with $n = 3$ results to be the most suitable choice in many cases, proving the validity of processing RGB images in their natural domain leveraging hypercomplex algebra. However, performance with $n = 3$ and $n = 4$ are comparable, thus the choice of this hyperparameter may depend on the application or on the hardware employed. On one hand, $n = 4$ may sometimes lead to lower performances, nevertheless it allows saving disk memory, as shown in Table 3, thus it may be more appropriate for edge applications. On the other hand, processing color images with $n = 3$ may bring higher accuracy even so it requires more parameters. Therefore, such a flexibility makes PHC models adaptable to a large range of applications. Likewise, PHC networks with $n = 2$ gain considerable accuracy scores with respect to the real-valued corresponding models and, due to the larger number of parameters with respect to the PHC with $n = 3$, sometimes outperform it too. Finally, the PHC with $n = 4$ obtains the overall best accuracy in the largest experiment of this table. Indeed, considering a ResNet152 backbone on CIFAR100, our method exceeds the real-valued baseline by more than $4\%$. This is the empirical proof that, with a very small FLOPs increase, PHC models well scale to large real-world problems by notably reducing the overall number of parameters. What is more, in Table 3, we report the memory required to store models checkpoints for inference. Our method crucially reduces the amount of disk memory demand with respect to the heavier real-valued model.

Further, we perform the image classifcation task on the ImageNet dataset. Since the most valuable choice for $n$ when dealing with RGB images has been proved to be 3, we test the real-valued ResNet50 against the PHC counterpart with $n = 3$. We train the models for 300k iterations with batch size 256 following the recipes in (Wightman et al., 2021). Table 4 shows that the proposed method achieves comparable, and even slightly superior, performance than the real-valued baseline, while involving 66% fewer parameters. This proves the robustness of the proposed PHC approach, which can be adopted and implemented in models at different scales.

## 4.2 HYPERCOMPLEX PARAMETERIZATION FOR SOUND EVENT DETECTION

Sound event detection (SED) is the task of recognizing the sounds classes and at what temporal instances these sounds are active in an audio signal (Mesaros et al., 2021). We prove that the PHC layer is adaptable to $n$-dimensional input signals and, due to parameters reduction and hypercomplex algebra, is more performing in terms of efficiency and evaluation scores.

Table 5: SEDnets results with one microphone (4 channels input). Scores are computed over three runs with different seeds and we report the mean. The proposed method wtih $n = 2$ far exceeds the baselines in each metric considered.

| Model | Conv Params | $F_{score} \uparrow$ | $ER \downarrow$ | $SED_{score} \downarrow$ | $P \uparrow$ | $R \uparrow$ |
|---|---|---|---|---|---|---|
| SEDnet | 1.6M | 0.637 | 0.450 | 0.406 | 0.756 | 0.5505 |
| Quaternion SEDnet | 0.4M (-75%) | 0.580 | 0.516 | 0.468 | 0.724 | 0.484 |
| PHC SEDnet $n = 2$ | 0.8M (-50%) | **0.680** | **0.389** | **0.355** | **0.767** | **0.611** |
| PHC SEDnet $n = 4$ | 0.4M (-75%) | 0.638 | 0.453 | 0.407 | 0.765 | 0.547 |

Table 6: SEDnets results with two microphones (8 channels input). Scores are computed over three runs with different seeds and we report the mean. The PHC SEDnet $n = 2$ outperform the baselines.

| Model | Conv Params | $F_{score} \uparrow$ | $ER \downarrow$ | $SED_{score} \downarrow$ | $P \uparrow$ | $R \uparrow$ |
|---|---|---|---|---|---|---|
| SEDnet | 1.6M | 0.663 | 0.428 | 0.383 | **0.788** | 0.572 |
| Quaternion SEDnet | 0.4M (-75%) | 0.559 | 0.556 | 0.499 | 0.754 | 0.444 |
| PHC SEDnet $n = 2$ | 0.8M (-50%) | **0.669** | **0.406** | **0.368** | 0.767 | **0.594** |
| PHC SEDnet $n = 4$ | 0.4M (-75%) | 0.638 | 0.433 | 0.397 | 0.729 | 0.567 |
| PHC SEDnet $n = 8$ | 0.2M (-87%) | 0.553 | 0.560 | 0.503 | 0.747 | 0.439 |

### 4.2.1 PARAMETERIZED HYPERCOMPLEX SEDNETS

Sound Event Detection networks (SEDnets) (Adavanne et al., 2019) are comprised of a core convolutional component which extracts features from the input spectrogram. The information is then passed to a gated recurrent unit (GRU) module and to a stack of fully connected (FC) layers with a closing sigmoid $\sigma$ which outputs the probability the sound is in the audio frame. Formally, the PHC SEDnet is described by

$$\begin{aligned} \mathbf{h}_t &= \text{PHC}_t(\mathbf{h}_{t-1}) \qquad t = 1, ..., j \\ \mathbf{y} &= \sigma\left(\text{FC}\left(\text{GRU}\left(\mathbf{h}_j\right)\right)\right). \end{aligned} \tag{8}$$

After the GRU model, We employ standard fully connected layers, that can be also implemented as PHM layers with $n = 1$, since the so processed signal loses its multidimensional original structure.

### 4.2.2 EXPERIMENTAL SETUP AND EVALUATION

For sound event detection models we consider the augmented version of the SELDnet (Adavanne et al., 2019; Comminiello et al., 2019) which was proposed as baseline for of the L3DAS21 Challenge Task 2 (Guizzo et al., 2021) and we perform our experiments with the corresponding released dataset[2]. We consider as our baselines the SEDnet (without the localization part) and its quaternion counterpart. The L3DAS21 Task 2 dataset contains 15 hours of MSMP B-format Ambisonics audio recordings, divided in 1-minute-long data points, each of which consists of a simulated 3D office audio environment where up to 3 acoustic events may overlap. We perform experiments with multiple configurations of this dataset. We first test the recordings from one microphone considering the magnitudes only (4 channels input), then we test the networks with the signals recorded by two microphones and magnitudes only (8 channels input). In Appendix B we report the experiments and the results up to 16 channels input, in which we consider the phase information too.

We investigate PHC SEDnets in complex, quaternion and octonion domain with $n = 2, 4, 8$ and train each network for 1000 epochs with a batch size of 16. Other hyperparameters are set as suggested in the paper (Guizzo et al., 2021) and more details are provided in Appendix B. To assess the performances of our models we take various metrics into account: the $F_{score}$ on the detection metric, Precision (P) and Recall (R) as suggested in the original paper. For a more rigorous evaluation, we also compute the $SED_{score}$ and the Error Rate (ER) (Adavanne et al., 2019; Mesaros et al., 2021).

---

[2]L3DAS21 dataset and code are available at: `https://github.com/l3das/L3DAS21`.

Table 7: SEDnets FLOPs, training (T) and inference (I) time with $8$ channels input. For training time (seconds/iteration) the mean and the standard deviation over one epoch is reported, for inference time we report the time required to perform an iteration on the validation set.

| Model | FLOPs | Time (T) | Time (I) |
|---|---|---|---|
| SEDnet | 37.3G | $1.242 \pm 0.088$ | 1.198 |
| Quaternion SEDnet | 37.3G | $1.308 \pm 0.088$ | 1.298 |
| PHC SEDnet $n = 2$ | 37.3G | $\mathbf{1.091 \pm 0.074}$ | 1.085 |
| PHC SEDnet $n = 4$ | 37.3G | $\mathbf{1.091 \pm 0.032}$ | **1.077** |
| PHC SEDnet $n = 8$ | 37.3G | $1.142 \pm 0.042$ | 1.173 |

The proposed parameterized SEDnets distinctly outperform real and quaternion-valued baselines, as reported in Table 5 and Table 6. Indeed, the PHC SEDnet with $n = 2$ gains the best results for each score and in both one and two microphone datasets, proving that the weights sharing due to the hypercomplex parameterization is able to capture more information regardless the lower number of parameters. It is interesting to note that the PHC SEDnet $n = 4$, which operates in the quaternion domain, achieves improved scores with respect to the Quaternion SEDnet that follows the rigid predefined algebra rules. Further, the malleability of PHC layers allows gaining comparable performance with respect to the quaternion baseline even so reducing convolutional parameters by $87\%$, just setting $n = 8$. In Appendix B, we show additional experimental results of PHC models able to save $94\%$ of convolutional parameters while operating in the sedonion domain by involving $n = 16$.

Furthermore, PHC SEDnets are more efficient in terms of time required for training and inference, regardless the equal number of FLOPs. Table 7 shows that each tested version of the proposed method is faster regards as the real SEDnet and the quaternion one, both at training and at inference time. Time efficiency is crucial in audio applications where networks are usually trained for thousands of epochs and datasets are very large and require protracted computations.

## 5 RELATED WORKS

While a research field deals with scaling up networks (Real et al., 2019), a consistent area in literature aims instead at making these models more efficient and accessible (Tan & Le, 2019; Sandler et al., 2018). Among the latter, quaternion neural networks (QNNs) have proved to reduce the amount of parameters to $1/4$ (Parcollet et al., 2019a; 2017). QNNs ensure high performance thanks to the sharing of weights due to the Hamilton product that preserves correlations among channels in multidimensional inputs (Grassucci et al., 2021c; Tay et al., 2019; Grassucci et al., 2021b). However, despite the lower number of parameters, QNNs are often slightly slow with respect to real-valued baselines (Hoffmann et al., 2020). Recently, several attempts have been made to compress neural networks relying on Kronecker product decomposition (Huang et al., 2020; Tang et al., 2021). These methods gain considerable results in terms of model efficiency (Wang et al., 2021). Lately, a parameterization of hypercomplex multiplications have been proposed to generalize hypercomplex fully connected layers by sum of Kronecker products (Zhang et al., 2021). The latter method obtains high performance in various natural language processing tasks by also reducing the number of overall parameters. Lately, other works extended this approach to graph neural networks (Le et al., 2021) and transfer learning (Mahabadi et al., 2021), proving the effectiveness of Kronecker product decomposition for hypercomplex operations.

## 6 CONCLUSION

In this paper, we introduce a parameterized hypercomplex convolutional (PHC) layer which grasps the convolution rule directly from data and can operate in any domain from 1D to $n$D, regardless the algebra regulations are preset. The proposed approach reduces the convolution parameters to $1/n$ with respect to real-valued counterparts and allows capturing internal latent relations thanks to parameter sharing among input dimensions. We show our method is flexible to operate in different fields of application by performing experiments with images and audio signals. We also prove the malleability and the robustness of our approach to learn convolution rules in any domain by setting different values for the hyperparameter $n$ from 2 to 16.

REPRODUCIBILITY STATEMENT

One of the objective of proposing methods that allow a reduction of parameters is to make large models more accessible and reproducible, even with a lower budget. We strongly believe reproducibility should be a key ingredient for recent deep learning papers. Therefore, we provide detailed explanations in the Appendix A to better understand the proposed method. Moreover, each experiment reported in the main corpus and in the appendix of this paper is completely reproducible. We provide the full code at `https://anonymous.4open.science/r/HyperNets-CBBB`, including models, train configurations, L3DAS21 preprocessing and information for dataset download and thorough instructions and details are provided in the Appendix B. Notebooks tutorials with toy examples are uploaded too, hoping these may help further research on this topic.

CO2 EMISSION RELATED TO EXPERIMENTS

Experiments were conducted using a private infrastructure, which has a carbon efficiency of 0.445 $kgCO_2eq/kWh$. A cumulative of 2000 hours of computation was performed on hardware of type Tesla V100-SXM2-32GB (TDP of 300W). Total emissions are estimated to be 267 $kgCO_2eq$ of which 0 percents were directly offset. Estimations were conducted using the MachineLearning Impact calculator presented in Lacoste et al. (2019).

More in detail, considering an experiment for the sound event detection (SED) task, according to Table 7, the real-valued baseline requires approximately 20 hours for training and validation, with a corresponding carbon emissions of 2.71 $kgCO_2eq$. Conversely, the proposed PHC model takes approximately 17 hours with a reduction of carbon emissions of 16%, being 2.28 $kgCO_2eq$.

In conclusion, we believe that the improved efficiency of our method with respect to standard models may be a little step towards reducing carbon emissions.

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

# A  THEORETICAL ADD-ONS

## A.1  QUATERNION ALGEBRA

A quaternion $q \in \mathbb{H}$ is an hypercomplex number of rank 4. It is defined as the composition of four real coefficients with three imaginary units as follows:

$$q = q_0 + q_1\hat{\imath} + q_2\hat{\jmath} + q_3\hat{\kappa} = q_0 + \mathbf{q}, \tag{9}$$

being $q_0$, $q_1$ $q_2$, $q_3 \in \mathbb{R}$ and $\hat{\imath}$, $\hat{\jmath}$, $\hat{\kappa}$ orthonormal basis in $\mathbb{R}^3$. A quaternion with scalar part $q_0$ equal to 0 is a *pure quaternion*. Considering two quaternion $q$ and $p$, the Hamilton product in Eq.2 can be rewritten in concise form as:

$$qp = q_0p_0 - \mathbf{q} \cdot \mathbf{p} + q_0\mathbf{p} + p_0\mathbf{q} + \mathbf{q} \times \mathbf{p},$$

where the bold notation is intended for vectors and the plain notation for scalars. From the above definition, it is easy to derive the Hamilton product between two pure quaternions. It takes the form:

$$\mathbf{qp} = -\mathbf{q} \cdot \mathbf{p} + \mathbf{q} \times \mathbf{p},$$

in which $-\mathbf{q} \cdot \mathbf{p}$ is the scalar part of the new quaternion and $\mathbf{q} \times \mathbf{p}$ its vector part. Therefore, the vector product between two pure quaternions gives rise to a full quaternion.

As complex numbers, also quaternions have their own complex conjugates that are defined as

$$q^* = q_0 - q_1\hat{\imath} - q_2\hat{\jmath} - q_3\hat{\kappa} = q_0 - \mathbf{q}.$$

The norm of a quaternion is the Euclidean norm in $\mathbb{R}^4$ of its real coefficients, that is

$$|q| = \sqrt{qq^*} = \sqrt{q_0^2 + q_1^2 + q_2^2 + q_3^2},$$

while the inverse is

$$q^{-1} = \frac{q^*}{|q|^2}.$$

## A.2  DEMYSTIFYING PHC LAYERS

We provide a formal explanation of the PHC layer to better understand the Kronecker product and how it organizes convolution filters to reduce the overall number of parameters to $1/n$. In Eq.10, we show how the PHC layer generalizes from 1D to $n$D domains. When subsuming real-valued convolutions in the first line of Eq.10, the Kronecker product is performed between a scalar $A$ and the filter matrix $\mathbf{F}$, whose dimension is the same as the final weight matrix $\mathbf{H}$, that is $s \times d \times k \times k$. Considering the complex case with $n = 2$ in the second line of Eq.10, the algebra is defined in

$\mathbf{A}_1$ and $\mathbf{A}_2$ while the filters are contained in $\mathbf{F}_1$ and $\mathbf{F}_2$, each of dimension $1/4$ the final matrix $\mathbf{H}$. Therefore, while the size of the weight matrix $\mathbf{H}$ remains unchanged, the parameter size is approximately $1/2$ the real one. In the last line of Eq.10, we can see the generalization of this process, in which the size of matrices $\mathbf{F}_i$, $i = 1, ..., n$ is reduced proportionally to $n$. It is worth noting that, while the parameter size is reduced with growing values of $n$, the dimension of $\mathbf{H}$ remains the same.

$$
\begin{aligned}
&\underset{(1\times1)}{[A]} \otimes \underset{(s\times d\times k\times k)}{\left[\ \mathbf{F}\ \right]} = \underset{(s\times d\times k\times k)}{\left[\ \mathbf{H}\ \right]} \\[2em]
&\underset{(2\times2)}{[\mathbf{A}_1]} \otimes \underset{\left(\frac{s}{2}\times\frac{d}{2}\times k\times k\right)}{\left[\ \mathbf{F}_1\ \right]} + \underset{(2\times2)}{[\mathbf{A}_2]} \otimes \underset{\left(\frac{s}{2}\times\frac{d}{2}\times k\times k\right)}{\left[\ \mathbf{F}_2\ \right]} = \underset{(s\times d\times k\times k)}{\left[\ \mathbf{H}\ \right]} \\[1em]
&\qquad\qquad\qquad\qquad\qquad \vdots \\
&\underset{(n\times n)}{[\mathbf{A}_1]} \otimes \underset{\left(\frac{s}{n}\times\frac{d}{n}\times k\times k\right)}{\left[\ \mathbf{F}_1\ \right]} + \underset{(n\times n)}{[\mathbf{A}_2]} \otimes \underset{\left(\frac{s}{n}\times\frac{d}{n}\times k\times k\right)}{\left[\ \mathbf{F}_2\ \right]} + \ldots + \underset{(n\times n)}{[\mathbf{A}_n]} \otimes \underset{\left(\frac{s}{n}\times\frac{d}{n}\times k\times k\right)}{\left[\ \mathbf{F}_n\ \right]} = \underset{(s\times d\times k\times k)}{\left[\ \mathbf{H}\ \right]}.
\end{aligned}
\tag{10}
$$

## B    EXPERIMENTAL ADD-ONS

### B.1    TOY EXAMPLES: LEARNING THE MATRIX $\mathbf{A}$

We test the receptive ability of the PHC layer in two toy problems building an artificial dataset. The first task is learning the proper matrix $\mathbf{A}$ to build a quaternion convolutional layer which properly follows the Hamilton rule in Eq.2. We build the dataset by performing a convolution with a matrix of filters $\mathbf{W} \in \mathbb{H}$, that are arranged following the regulation in Eq.2, and a quaternion $\mathbf{x} \in \mathbb{H}$ in input. The target is still a quaternion, named $\mathbf{y} \in \mathbb{H}$. As shown in Fig. 2 (right), the PHC loss converges very fast, meaning that the layer properly learns the matrix $\mathbf{A}$ and the Hamilton convolution.

The second toy example is a modification of the previous dataset target.. Here, we want to learn the matrix $\mathbf{A}$ which describes the convolution among two pure quaternions. Pure quaternions may be, as an example, an input RGB image and the weights of a hypercomplex convolutional layer. Figure 2 (left) displays the convergence of the PHC layer loss during training, proving that the proposed method is able of subsuming hypercomplex convolutional rules when dealing with pure quaternions too..

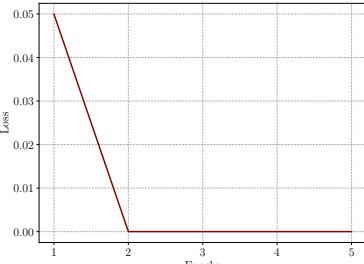 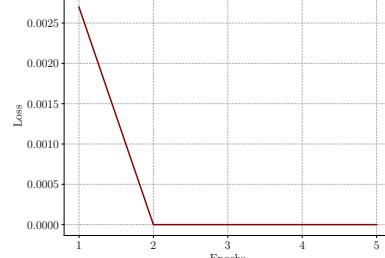

Figure 2: Loss plots for toy examples. The PHC layer is able to learn the matrix **A** which describes the convolution rule for pure quaternions (left) and for full quaternions (right).

Table 8: VGG16 results with real-valued classifier for quaternion and PHC networks. Extension of Table 1 in the main corpus.

| Model | Params | SVHN | CIFAR10 |
|---|---|---|---|
| Quaternion VGG16 | 4.2M (-72%) | 94.086 | 84.126 |
| PHC VGG16 $n = 2$ | 7.9M (-62%) | **94.885** | **86.147** |
| PHC VGG16 $n = 4$ | 4.2M (-72%) | 94.562 | 85.710 |

## B.2 EXPERIMENTAL ADD-ONS ON IMAGE CLASSIFICATION

### B.2.1 IMAGE CLASSIFICATION IMPLEMENTATION DETAILS

To guarantee the complete reproducibility of our experiments, here we provide more implementation details for the image classification task.

The modified versions of the ResNets are built with an initial convolutional layer with $60$ filters. Then, the subsequent blocks have $60, 120, 240, 516$ filters. The number of layers in the blocks depends on the ResNet chosen, whether 18, 50 or 152.

Instead, VGG19 convolution component comprise two 24, two 72, four 216, and eight 648 filter layers, with batch normalization. The classifier is composed of three fully connected layers of 648, 516 and with the number of classes neurons each.

The rest of the hyperparameters are set as suggested in the original papers. The batch size is fixed to $128$ and training is performed via SGD optimizer with momentum equal to $0.9$, weight decay $5e^{-4}$ and a cosine annealing scheduler. For ResNets, the initial learning rate is set to $0.1$. For VGG is equal to $0.01$. Models on CIFAR10 and CIFAR100 are trained for 200 epochs whereas on SVHN networks run for 50 epochs.

For the ImageNet dataset, we follow the recipes in (Wightman et al., 2021), so we resize the images for training at $160 \times 160$ while keeping the standard size of $224 \times 224$ for validation and test. We employ a step learning rate decay every 30 epochs with $\gamma = 0.1$, the SGD optimizer and an initial learning rate of $0.1$ with weight decay $0.0001$. The training is performed for 300k iterations with a batch size of 256 employing 4 GPUs.

### B.2.2 ADDITIONAL EXPERIMENTS ON IMAGE CLASSIFICATION

In the following, we provide additional preliminary experiments with small models equipped with the proposed PHC layer. Table 8 contains VGG16s with real-valued final classifier. Table 9 reports experiments with ResNet20 where we test also $n = 1$ to replicate the real-valued model, outperforming it. Experiments with VGG11 with modified number of filters in order to be divided by each value of $n$ is also reported in the same table. Finally, in Table 10 additional experiment on SVHN and CIFAR10 with ResNet56 and ResNet110, the latter with modified number of filters. PHC-based

Table 9: Additional preliminary experiments on SVHN dataset with ResNet20 and VGG11, the latter with modified number of filters in order to be divided by each value of $n$ and FC layers in the closing classifier. We test also the PHC model with $n = 1$ to replicate the real domain which outperform the real-valued ResNet20.

| Model | Params | SVHN |
|---|---|---|
| ResNet20 | 0.27M | 90.463 |
| Quaternion ResNet20 | 0.07M (-75%) | 93.535 |
| PHC ResNet20 $n = 1$ | 0.27M | **93.7962** |
| PHC ResNet20 $n = 2$ | 0.14M (-50%) | 93.708 |
| PHC ResNet20 $n = 4$ | 0.07M (-75%) | 93.669 |
| VGG11 | 13.8M | 93.488 |
| Quaternion VGG11 | 3.9M (-71%) | 92.888 |
| PHC VGG11 $n = 2$ | 7.2M (-48%) | **93.958** |
| PHC VGG11 $n = 3$ | 5.0M (-64%) | 93.804 |
| PHC VGG11 $n = 4$ | 3.9M (-71%) | 93.919 |

Table 10: Additional preliminary experiments with ResNet56 and ResNet110, the latter with modified number of filters in order to be divided by each value of $n$. Accuracy score is the mean over three runs with different seeds.

| Model | Params | SVHN | CIFAR10 |
|---|---|---|---|
| ResNet56 | 0.9M | 94.116 | **83.700** |
| Quaternion ResNet56 | 0.2M (-75%) | 93.664 | 81.687 |
| PHC ResNet56 $n = 2$ | 0.4M (-50%) | 93.722 | 83.413 |
| PHC ResNet56 $n = 4$ | 0.2 (-75%) | **94.122** | 82.720 |
| ResNet110 | 16.7M | 93.461 | 84.810 |
| Quaternion ResNet110 | 4.2M (-75%) | 92.788 | 83.920 |
| PHC ResNet110 $n = 2$ | 8.4M (-50%) | 93.746 | 83.220 |
| PHC ResNet110 $n = 3$ | 5.6M (-66%) | 94.712 | 85.200 |
| PHC ResNet110 $n = 4$ | 4.2M (-75%) | **94.885** | **85.280** |

Table 11: Additional experiments with ResNet-based models. We reduced the number of convolutional filters by $75\%$ and then test the models on three datasets to remove the hypothesis that a smaller number of parameters leads to higher generalization capabilities.

| Model | Params | SVHN | CIFAR10 | CIFAR100 |
|---|---|---|---|---|
| ResNet18 | 10.1M | **93.992** | **89.543** | **62.634** |
| ResNet18 (reduced) | 2.7M (-75%) | 93.842 | 88.310 | 59.590 |
| ResNet50 | 22.5M | **94.546** | **89.630** | **65.514** |
| ResNet50 (reduced) | 5.7M (-75%) | 93.915 | 89.370 | 62.450 |
| ResNet152 | 52.6M | **94.625** | **89.580** | **62.053** |
| ResNet152 (reduced) | 13.2M (-75%) | 94.400 | 89.001 | 60.850 |

models gain good performance in each test we conduct while reducing the amount of free parameters. Finally, in order to further remove the hypothesis that smaller number of neural parameters leads to higher generalization capabilities, we perform experiments with real-valued baselines with a number of parameters reduced by $75\%$. Table 11 shows that reducing the number of filters downgrades the performance and thus it is not sufficient to improve the generalization capabilities of a model.

### B.3 EXPERIMENTAL ADD-ONS ON SOUND EVENT DETECTION

### B.3.1 L3DAS21 DATASET DETAILS

The L3DAS21 Task 2 dataset is a collection of 900 1-minute-long data-points sampled at a rate of 32 kHz. The 14 sounds classes have been selected from the FSD50K dataset and are representative for an office sounds: computer keyboard, drawer open/close, cupboard open/close, finger snapping, keys jangling, knock, laughter, scissors, telephone, writing, chink and clink, printer, female speech, male speech. In this dataset, the volume difference between the sounds is in the range 0 and 20 dB full scale (dBFS). Considering the array of two microphones $1, 2$, the channels order is [W1, Z1, Y1, X1, W2, Z2, Y2, X2], where WXYZ are the B-format ambisonics channels if the phase (p) information is not considered. Whether we want to include also this information, the order will be [W1, Z1, Y1, X1, W1p, Z1p, Y1p, X1p, W2, Z2, Y2, X2, W2p, Z2p, Y2p, X2p] up to 16 channels. In Fig.3, we show the 8-channel input when considering one microphone and the phase information. Magnitudes and phases are normalized to be centered in 0 with standard deviation 1.

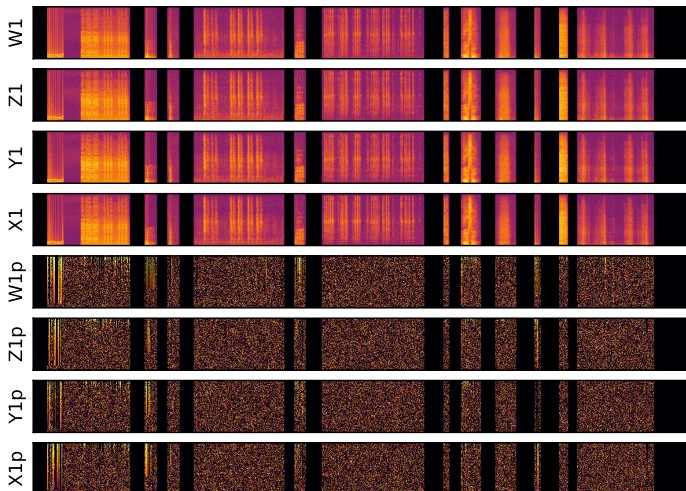

Figure 3: Sample spectrograms from L3DAS21 dataset recorded by one microphone with four capsules.The first four figures represent the magnitudes while the last four contain the corresponding phases information. The black sections represent silent instants.

### B.3.2 SED IMPLEMENTATION DETAILS

To guarantee the complete reproducibility of our experiments, here we provide more implementation details for the sound event detection task.

The extracted features by the preprocessing are fed to the four-layer convolutional stack with $64, 128, 256, 512$ filters, with batch normalization, ReLU activation, max pooling and dropout (probability $0.3$), with pooling sizes $(8, 2), (8, 2), (2, 2), (1, 1)$. The bidirectional GRU module has three layers, each with an hidden size of $256$. The tail has is a four-layer fully connected classifier with $1024$ filters alternated by ReLUs and with a final dropout and a sigmoid activation function. The initial learning rate is set to $0.00001$.

Following, we define the equations of the evaluation metrics considered in Subsection 4.2. To be consistent with pre-existing literature, we define True Positives as TP, False Positives as FP and False Negatives as FN. These are computed according to the detection metric (Guizzo et al., 2021). Moreover, in order to compute the Error Rate (ER), we consider: $S = \min(FN, FP)$, $D = \max(0, FN - FP)$ and $I = \max(0, FP - FN)$, as in (Adavanne et al., 2019).

$$F_{\text{score}} = \frac{2TP}{2TP + FP + FN}$$

$$ER = \frac{S + D + I}{N}$$

whereby $N$ is the total number of active sound event classes in the reference.

$$SED_{\text{score}} = \frac{ER + 1 - F_{\text{score}}}{2}$$

For ER and $SED_{\text{score}}$, the lower scores, the better the performances, while for the $F_{\text{score}}$ higher values stand for better accuracy.

### B.3.3 ADDITIONAL EXPERIMENTS ON SOUND EVENT DETECTION

In this section, we provide additional experiments for the sound event detection task. We conduct a test considering two microphones and the phase information, so to have an input with 16 channels. For this purposes, we consider as baseline the quaternion model and PHC-based networks with $n = 4, 8, 16$ so to test higher order domains. Results are shown in Table 12.

Table 12: SED results with two microphone: magnitudes and phases (16 channels input). We test higher order hypercomplex domains up to sedonions by setting $n = 16$. Although the incredible reduction of the number of parameters with respect to the real-valued baseline in Table 6, the PHC with $n = 16$ still has comparable performances with other models. Furthermore, the PHC with $n = 8$ outperform also the quaternion baseline which has more degrees of freedom.

| Model | Conv Params | $F_{\text{score}} \uparrow$ | ER $\downarrow$ | $SED_{\text{score}} \downarrow$ | P $\uparrow$ | R $\uparrow$ |
|---|---|---|---|---|---|---|
| Quaternion SEDnet | 0.4M (-75%) | 0.580 | 0.480 | 0.450 | 0.655 | 0.520 |
| PHC SEDnet $n = 4$ | 0.4M (-75%) | 0.585 | 0.470 | 0.443 | 0.653 | 0.530 |
| PHC SEDnet $n = 8$ | 0.2M (-87%) | **0.607** | **0.466** | **0.430** | 0.702 | **0.534** |
| PHC SEDnet $n = 16$ | 0.1M (-94%) | 0.588 | 0.509 | 0.461 | **0.734** | 0.491 |

## C FUTURE PERSPECTIVES

In this work, we demonstrate that the proposed parameterized hypercomplex convolutional (PHC) layer is able to consistently reduce the free parameters of neural models while ensuring very high performances thanks to the hypercomplex convolution properties. Therefore, we believe that these findings pave the way for novel efficient neural networks. As an example, a thorough study on pruning PHC-based models may further reduce the number of parameters and lead to extremely lightweight networks.

