# OpenReview forum: "Lightweight Convolutional Neural Networks By Hypercomplex Parameterization"
_ICLR.cc/2022/Conference — ICLR 2022 Submitted_

### Official Review · Reviewer_KMob · 2021-10-25

**Correctness:** 3
**Technical Novelty And Significance:** 3
**Empirical Novelty And Significance:** 3
**Recommendation:** 8
**Confidence:** 5

**Main Review:**

Summarising this paper is quite straightforward as the contribution is quite simple, yet effective. The concept of hypercomplex parameterization is extended to convolutional layers and common CNN models are also turned into hypercomplex parameterized models. The new model performs better than all baselines with less neural parameters and slightly (always ?) quicker training and inference times compared to hypercomplex models. In practice, the contribution is minimal, but very valuable as the code is given. If the code is open to the community afterwards, it is a win-win situation. Simple, efficient, and works well.

Seing this concept of hypercomplex parameterization extended is very nice considering that is has changed the efficiency capabilities of hypercomplex networks. Indeed, despite promising results and memory footprint, hypercomplex networks suffered from a lack of generalisation AND a crazy training / inference time compared to real-valued neural networks. This is all almost solved with the hypercomplex parameterization introduced in a previous paper and extended to CNN in this work.

My only concern, that prevents me from assigning a high rank on the current submission relates to the quality of the writing. I will only provide two examples, but the paper should really be proofread carefully and the English must be improved to meet ICLR requirements. If this is done properly, I don't see why this paper shouldn't be accepted.
"But the layers saves the 75% of free parameters" / "that consider deep layers [...], thus [...]"

An other smaller comment concerns Section 3.1. It is really weird to have results presented in formal Section. I would encourage the authors to move this to the experimental part instead, or rewrite the two last paragraphs as they currently sound as "look, we tried, it works, but we don't show you the results" even if you give then in Appendix. It is not appropriate to present it in this way and at this stage of the paper.

**Strengths:**
- Simple idea, yet very effective.
- Definitely impactful for hypercomplex neural network (and thus the concerned community).
- If the code is given, it will be very helpful to the community.
- Good results as expected (since they were already observed with non-convolutional layers).

**Weaknesses:**
- Writing must be polished.
- Add time and inference for ALL tables as this is a major point in the discussion between hypercomplex networks and parameterised ones !
- Add real-valued models with the exact same number of neural parameters than PHC ones to the tables. This will remove the hypothesis that smaller number of neural parameters leads to higher generalisation capabilities.


**Summary Of The Paper:**

This work extends the recently proposed hypercomplex parameterization of neural layers to convolutional layers. Then it describe common deep CNN models (ResNet, VGG, SeDnet) based on this novel paradigm in two different tasks (Image Classification and Sound Event Detection). The introduced method offer better performance with a reduced number of neural parameters as well as faster training and inference than equivalent hypercomplex layers.

**Summary Of The Review:**

Simple and impactful idea. Results are convincing. It will be an interesting paper to the hypercomplex networks community. Nevertheless, writing must be improved to get accepted (not the outline of the paper but the content).

---

> ### Author Response · Authors · 2021-11-22
> **On Section 3.1**
>
> We would like to thank the Reviewer for her/his suggestion. We have revised the paper as suggested.
> Since the plots would not have entered the main body, also due to new material added in the revised paper, we have moved the whole experiment from Section 3.1 to Appendix B.2.1, where it can be found now in a complete form, including also the plots of the losses. We have also referred to this experiment in the main body.

---

> > ### Comment · Reviewer_KMob · 2021-11-24
> > **Thanks for the work**
> >
> > I would like to acknowledge the work done on this rebuttal and on this submission. I am now leaning towards acceptance, I'll increase the score.

---

> > > ### Author Response · Authors · 2021-11-29
> > > **Thanks for the positive feedback**
> > >
> > > We would sincerely thank the Reviewer for her/his positive feedbacks and for the score increase.

---

> ### Author Response · Authors · 2021-11-22
> **On training time**
>
> We would like to thank the Reviewer for this suggestion. As suggested by the Reviewer, we have computed the training and inference time for the models we tested. We report a summary of the results in the table below. However, since the results are similar to the already computed ones for VGGs and do not add any additional claim, we believe that adding this information to the main body may result in redundant information and make the table a bit heavy to be read.
> However, to be as clear as possible, we specify this information in the revised version of the paper (see Table 2 caption).
> Training and inference time for ResNet-based models.
>
> | Model               | Training Time (s/it) | Inference Time (s) |
> |---------------------|----------------------|--------------------|
> | ResNet18            | 0.019 +- 0.019       | 0.003              |
> | Quaternion ResNet18 | 0.034 +- 0.022       | 0.005              |
> | PHC ResNet18 n=2    | 0.032 +- 0.022       | 0.006              |
> | PHC ResNet50 n=3    | 0.027 +- 0.017       | 0.005              |
> | PHC ResNet152 n=4   | 0.027 +- 0.016       | 0.005              |
> | ResNet50            | 0.079 +- 0.017       | 0.006              |
> | Quaternion Resnet50 | 0.142 +- 0.018       | 0.013              |
> | PHC ResNet50 n=2    | 0.121 +- 0.017       | 0.012              |
> | PHC ResNet50 n=3    | 0.122 +- 0.015       | 0.011              |
> | PHC ResNet50 n=4    | 0.122 +- 0.017       | 0.012              |

---

> ### Author Response · Authors · 2021-11-22
> **On generalization with lower parameters**
>
> We would like to thank the Reviewer for this suggestion. To be consistent with the literature, we tested VGG16, VGG19, Resnet18, ResNet50, and ResNet152 in the main body. However, in Appendix B.2.2 we reported the experiments with smaller networks (VGG11, ResNet20, ResNet56, and ResNet110) for SVHN and CIFAR10. In the original ResNet paper (https://arxiv.org/pdf/1512.03385.pdf), these ResNets were specifically built for these datasets. However, despite the lower number of parameters of these models, our approach outperforms the baselines in this case too, removing the hypothesis that a smaller number of parameters may lead to a high generalization.
> To further claim this result, as suggested by the Reviewer, we have performed additional experiments reducing by 75% the number of filters of the real-valued ResNets that we initially considered as baselines in the main body. We have added the results in a table in the revised version of the paper (Table 11 in Appendix B.2.2) and we also report a summary below. This shows that reducing the number of parameters in real-valued models leads to a performance decrease, thus removing the hypothesis that lower parameters bring higher generalization.
> Here are some results from the reduced real-valued models (-75% of free parameters). In the table below, the new contribution is represented by the reduced models.
>
> | Model               | # Params     | SVHN   | CIFAR10 | CIFAR100 |
> |---------------------|--------------|--------|---------|----------|
> | ResNet18            | 10.1M        | 93.992 | 89.543  | 62.634   |
> | ResNet18 (reduced)  | 2.7M (-75%)  | 93.842 | 88.310  | 59.590   |
> | ResNet50            | 22.5M        | 94.546 | 89.630  | 65.514   |
> | ResNet50 (reduced)  | 5.7M (-75%)  | 93.915 | 89.370  | 62.450   |
> | ResNet152           | 52.6M        | 94.625 | 89.580  | 62.053   |
> | ResNet152 (reduced) | 13.2M (-75%) | 94.400 | 89.001  | 60.850   |

---

> ### Author Response · Authors · 2021-11-22
> **On the writing**
>
> We would like to thank the Reviewer for her/his positive feedback and her/his suggestions.
> We revised the paper and did our best to improve the writing quality. We hope that now the paper meets the ICLR requirements.

---

### Official Review · Reviewer_MVqX · 2021-11-02

**Correctness:** 4
**Technical Novelty And Significance:** 2
**Empirical Novelty And Significance:** 3
**Recommendation:** 6
**Confidence:** 4

**Main Review:**

Strength:

1. This method significantly improves the time and computational power required for the training of large neural networks while achieving good performance.
2. The method can be operated in domains from $1D$ to $nD$, where $n$ can be tuned to attain appropriate results on given data.
3. Neural models redefined with this approach require $1/n$ of free parameters compared to their real-valued counterparts.

Weakness:

1. Although the research has many real-world applications, it is lacking novelty. The paper is an extension of (Zhang et al., 2021) for the convolution case.
2. Many details required to reimplement these models to achieve similar results are missing in the paper.
3. The choice of $n$ should also be analyzed using FLOPs.

**Summary Of The Paper:**

The paper proposes Parametrized Hypercomplex convolutional Layers to replace convolutional layers to attain similar performance using fewer parameters. The PHC layer can subsume hypercomplex convolution rules. The main advantage of the PHC Layer is that they enable choosing an arbitrary $n$ to reduce the number of parameters to $1/n$, whereas this was limited to $4$, $8$, and $16$ with quaternions. Hyperparameter $n$ also drives the number of convolutional filters. The proposed network can represent a real-valued convolutional layer by setting $n = 1$.

The PHC layer differs from the convolutional layer in the way that the weight matrix is obtained. $H$ acts as a weight for input $x$. The $A$ and $F$ matrices which are trainable and used to obtain $H$. Kronecker products of every matrix in the tensors $A$ and $F$ are taken and then these are summed to get $H$.

Authors have proposed PHC versions of some common neural networks such as ResNet and VGG and have compared the results with real-valued models, Quaternion models, and PHC models (for $n=2$ and $n=4$).

**Summary Of The Review:**

The paper lacks novelty and is an extension of (Zhang et al., 2021) for the convolution case. This is a major concern for me.

---

> ### Author Response · Authors · 2021-11-22
> **On the experimental details**
>
> We understand the points raised by the Reviewer, however, many of these points were addressed in the Appendices. Due to the page limit, in the original version of the paper we provided thorough details about the implementations in Appendix B, including model details, hyperparameters, and dataset preprocessing. Moreover, we included the link for the repository of the experiments (anonymized for blind reviews purposes). In the repository, setup for the experiments, models, and training procedures to reproduce every experiment can be found. Furthermore, we also provided notebook tutorials to better understand our approach. At the end of the review process, we will release the official GitHub repository link. We repost the anonymized repository here: https://anonymous.4open.science/r/HyperNets-CBBB.
> We are aware that probably we did not always reference the Appendices in the main body. Thus, in the revised version, we have added more references to the Appendix and repository link in the paper in order to be as clear as possible.
> We thank the Reviewer for this comment.

---

> ### Author Response · Authors · 2021-11-22
> **On the novelty**
>
> We defined novel convolutional layers based on the sum of Kronecker products and we tested them with n=2,3,4 in Table 2, and up to n=8,16 in Table 6 and in Table 12. We also devised an innovative method to process color images in their natural domain but leveraging hypercomplex algebra properties. To the best of our knowledge, this is the first approach that processes color images via hypercomplex algebra in neural networks without adding any useless padding channel or information. From the results in Table 2, it is clear that our approach outperforms pre-existing methods in which additional information is attached to the input. Moreover, the same approach allows processing multichannel audio signals exploiting correlations among each of the 16 input channels. To the best of our knowledge, the proposed PHC is the first method that allows processing multichannel audio spectrograms grasping relations among each channel of an array of two Ambisonics microphones. Moreover, our approach is capable of reducing the convolutional parameters up to 1/16 (-94%) in our tests, while quaternion neural networks reach just 1/4 (-75%). This boosts the inference time and drastically saves storage memory, features that may be crucial in embedded applications.
> Finally, we proved that the proposed method is flexible to operate in different domains of application without any theoretical as well implementation modifications, while previous works focus on just one domain of application.
> We are aware that probably we did not give enough evidence to the above novel contributions in the paper, thus in the revised version of the paper, we have better highlighted novelty in the Introduction. We thank the Reviewer for pointing this out.

---

> ### Author Response · Authors · 2021-11-22
> **On the choice of n**
>
> We would like to thank the Reviewer for her/his brilliant suggestion.
> We computed MACs and FLOPs for SEDnet-based models and for any ResNet-based PHC model and we added the results in the paper (Table 2 and Table 7) and in the table below. Unfortunately, the FLOPs difference among the various $n$ values is very slight, almost unimportant. This is because the final weight matrix H that we build by PHC is of the same dimension as the real-valued one. However, we believe that FLOPs count may increase the information in the paper and may help clarify aspects of the implementation.
>
> Table for SEDnets
>
> | Model             | FLOPs  | MACs    |
> |-------------------|--------|---------|
> | SEDnet            | 37.3 G | 18.65 G |
> | Quaternion SEDnet | 37.3 G | 18.65 G |
> | PHC SEDnet n=2    | 37.3 G | 18.65 G |
> | PHC SEDnet n=4    | 37.3 G | 18.65 G |
> | PHC SEDnet n=8    | 37.3 G | 18.65 G |
>
>
> Table for ResNets
>
> | Model                | FLOPs  | MACs     |
> |----------------------|--------|----------|
> | ResNet18             | 1.01 G | 507.38 M |
> | Quaternion ResNet18  | 1.01 G | 507.38 M |
> | PHC ResNet18 n=2     | 1.03 G | 517.50 M |
> | PHC ResNet18 n=3     | 1.03 G | 518.24 M |
> | PHC ResNet18 n=4     | 1.03 G | 518.24 M |
> | ResNet50             | 2.36 G | 1.18 G   |
> | Quaternion ResNet50  | 2.36 G | 1.18 G   |
> | PHC ResNet50 n=2     | 2.41 G | 1.20 G   |
> | PHC ResNet50 n=3     | 2.41 G | 1.21 G   |
> | PHC ResNet50 n=4     | 2.41 G | 1.21 G   |
> | ResNet152            | 6.62 G | 3.31 G   |
> | Quaternion ResNet152 | 6.62 G | 3.31 G   |
> | PHC ResNet152 n=2    | 6.76 G | 3.38 G   |
> | PHC ResNet152 n=3    | 6.76 G | 3.38 G   |
> | PHC ResNet152 n=4    | 6.76 G | 3.38 G   |
>
> We would like to thank the Reviewer for her/his brilliant suggestion.
> We computed MACs and FLOPs for SEDnet-based models and for any ResNet-based PHC model and we added the results in the paper (Table 2 and Table 7) and in the tables here. Unfortunately, the FLOPs difference among the various $n$ values is very slight, almost unimportant. This is because the final weight matrix H that we build by PHC is of the same dimension as the real-valued one. However, we believe that FLOPs count may increase the information in the paper and may help clarify aspects of the implementation.

---

### Official Review · Reviewer_JhvG · 2021-11-03

**Correctness:** 3
**Technical Novelty And Significance:** 2
**Empirical Novelty And Significance:** 2
**Recommendation:** 5
**Confidence:** 4

**Main Review:**

Strengths:
* Results are shown on multiple domains and strong performance is achieved.
* The paper is clearly written.
* The authors provide code making the results easily reproducible by future groups.

Weaknesses:
* I think the experimental results could be fleshed out. The networks and datasets are fairly small. It would be interesting to see the results pushed to a greater scale. It would be nice to see, for example, some results with a ResNet50 on ImageNet.
* It isn't immediately clear to me that parameter count is what needs to be cared about the most. Especially with quaternion, there is a substantial FLOP increase. I think comparisons on FLOPs would be most interesting. If the argument is model size, quaternions will still induce an increase activation memory.


Comments/Questions:
* From my understanding of Deep Complex Networks from Trabelsi, and Deep Quaternion Networks from Gaudet, they argue that a whitening procedure should be used instead of BatchNorm. In AlgebraNets (https://arxiv.org/abs/2006.07360) the authors use BatchNorm, arguing the FLOP expense of whitening is not worth it. Is this something that you experimented with?  Similarly, did you consider other algebras, such as matrix rings?
* Deep Complex Networks, Deep Quaternion Networks, and AlgebraNets all investigate various activation functions beyond ReLU. Is this something that you considered?
* Other forms of model compression methods such as pruning are quite popular. Have you tried pruning any of these networks for further parameter reductions?


**Summary Of The Paper:**

Traditionally, networks use real-valued weights. However recent work has shown that in certain domains, weights that obey different algebraic rules can lead to various performance enhancements. The authors introduce a hyper complex convolution block and show that it leads to performance gains and parameter reductions over a variety of baseline models on tasks in various domains.

**Summary Of The Review:**

The authors propose an interesting idea, however I think that the presented comparisons miss a few key comparisons. Specifically, I would have liked to see a measure of FLOPs required to reach given accuracies. Additionally, while the results shown span multiple domains I would have liked to see more ambitious experiments.

---

> ### Author Response · Authors · 2021-11-22
> **On the large-scale experiments**
>
> We would like to thank the Reviewer for her/his positive feedback and suggestions to improve the paper.
> In the original version of the paper, we considered benchmark datasets at different scales for image classification, ranging from SVHN up to CIFAR100. Thereafter, instead of using ImageNet for a larger-scale experiment, we have considered another large-scale dataset, the L3DAS21 dataset, in order to underline the robustness of our approach on different scales as well as on different domains.
> Indeed, the L3DAS21 is one of the most difficult and large datasets available for this task. It contains 1-minute-long data points while other datasets consider shorter segments that imply an easier task. The TUT dataset (https://homepages.tuni.fi/annamaria.mesaros/pubs/mesaros_eusipco2016-dcase.pdf) implies 30 seconds segments, the TAU dataset (https://arxiv.org/pdf/1807.09840v2.pdf) implies 10 seconds segments, among others. Moreover, in the L3DAS21 dataset, sounds from the same class may overlap while this does not happen in other datasets. Additionally, the L3DAS21 dataset is recorded with two microphones, therefore for each data point, we can have up to 16 channels as expounded in Appendix B.3.1.
> However, as suggested by the Reviewer, we also perform an additional experiment on the ImageNet dataset. We added the results in a table in the revised version of the paper (see Table 4) and we report a summary below. We chose to perform the experiment with the PHC n=3 since it has been proved to be the most valuable choice when dealing with RGB images since it does not add any useless information to the input.  Results show that the proposed method achieves comparable, and even slightly superior, performance than the real-valued baseline, while involving 66% fewer parameters. This proves the robustness of the proposed PHC approach, which can be adopted and implemented in models at different scales.
> We would like to thank the Reviewer for this suggestion that has improved the robustness of our work.
>
> | Model            | # Params    | ImageNet |
> |------------------|-------------|----------|
> | ResNet50         | 25.7M       | 67.99    |
> | PHC ResNet50 n=3 | 9.6M (-66%) | 68.584   |

---

> > ### Comment · Reviewer_JhvG · 2021-11-30
> > **Thanks for your comments**
> >
> > The author(s) comments have helped clarify many aspects of the work. One question I still have is with regards to the activation memory-- while I understand quaternion approaches reduce the parameter count, it seems that activation memory would be increased-- is this the case?
> >
> > I have raised my score to a 5 as I still think that, though improved, a tightening of the narrative will greatly strengthen the paper.

---

> > > ### Author Response · Authors · 2021-11-30
> > > **On the memory efficiency**
> > >
> > > We would like to thank the Reviewer for his/her score, although we are sorry to know that our efforts did not go towards obtaining a sufficient evaluation.
> > >
> > > As the Reviewer rightly deduces, parameter efficiency does not necessarily imply memory efficiency. This is the case with quaternion and parameterized hypercomplex (PH) models. In our PH model, the memory occupation is certainly lower than the classic quaternion models but still slightly higher than the real-valued models.
> > > This aspect in our opinion is very crucial and will be one of the points we will work on in the future because obtaining memory efficiency would mean that PH methods could represent a valid alternative for devices with limited resources. This would also imply the redefinition of normalization methods in the hypercomplex domain, avoiding the drawbacks described also in the literature for quaternion-valued networks. However, this was not the main focus of this paper.
> > >
> > > We know that a tightening narrative on this aspect would certainly have been interesting an we are sorry to have not included it, but in 9 pages we did our best trying to present different theoretical aspects, as well as implementation and experimental aspects, comparing several PH models at different scales and in different application domains. We would have taken away from something else that is now in the paper, and it would have been penalizing anyway.
> > >
> > > Nonetheless, we would like to thank again the Reviewer for his/her comments because they have been inspiring and constructive for our work and have certainly touched on points that we will deepen in the near future.

---

> ### Author Response · Authors · 2021-11-22
> **On FLOPs comment**
>
> We would like to thank the Reviewer for the suggestion of considering FLOPs.
> We computed MACs and FLOPs for each ResNets we tested, we added FLOPs information in the revised version of the paper (Table 2 and Table 7) and we report a summary table here. Although in the paper AlgebraNets the authors claim that quaternion networks have a substantial FLOPs increase, we did not notice this increment. Unfortunately, MACs and FLOPs metrics strongly depend on the implementation and AlgebraNets authors did not release any official code, so we cannot check their implementation. Instead, for quaternion baselines, we employ the well-known repository by T. Parcollet (https://github.com/TParcollet/Quaternion-Neural-Networks) and we release the code for real, quaternion and PHC models in the repository (anonymized until the end of the review process) at this link: https://anonymous.4open.science/r/HyperNets-CBBB. Therefore, whoever can validate our results.
> We computed the count via the DeepSpeed tool (https://www.deepspeed.ai/) and we report the results in the table below.
> Our approach has a slightly higher number of FLOPs with respect to the real-valued baselines. This is because the final weight matrix H that we build by PHC is of the same dimension as the real-valued one. Thus, the FLOPs are equal to standard layers. The FLOPs increase is very slow, especially for SEDnets that already have a high number of floating point operations and include few PHC layers. Indeed, the small increase of our method is not significant in the overall FLOPs count:
>
> | Model             | FLOPs  | MACs    |
> |-------------------|--------|---------|
> | SEDnet            | 37.3 G | 18.65 G |
> | Quaternion SEDnet | 37.3 G | 18.65 G |
> | PHC SEDnet n=2    | 37.3 G | 18.65 G |
> | PHC SEDnet n=4    | 37.3 G | 18.65 G |
> | PHC SEDnet n=8    | 37.3 G | 18.65 G |
>
> However, we believe that these metrics should be jointly considered with training and inference time, since concatenation, reshaping and other operations like those do not affect FLOPs count but they affect model speed. Indeed, while quaternion baselines have the same FLOPs as real-valued models, they are far slower with respect to our method too. This is caused by heavy concatenation operations in the quaternion layers implementation. On the contrary, the implementation of PHC layers leverages a few unsqueeze operations for a faster PyTorch implementation. The table below reports the FLOPs and MACs count for ResNet-based models.
>
> | Model                | FLOPs  | MACs     |
> |----------------------|--------|----------|
> | ResNet18             | 1.01 G | 507.38 M |
> | Quaternion ResNet18  | 1.01 G | 507.38 M |
> | PHC ResNet18 n=2     | 1.03 G | 517.50 M |
> | PHC ResNet18 n=3     | 1.03 G | 518.24 M |
> | PHC ResNet18 n=4     | 1.03 G | 518.24 M |
> | ResNet50             | 2.36 G | 1.18 G   |
> | Quaternion ResNet50  | 2.36 G | 1.18 G   |
> | PHC ResNet50 n=2     | 2.41 G | 1.20 G   |
> | PHC ResNet50 n=3     | 2.41 G | 1.21 G   |
> | PHC ResNet50 n=4     | 2.41 G | 1.21 G   |
> | ResNet152            | 6.62 G | 3.31 G   |
> | Quaternion ResNet152 | 6.62 G | 3.31 G   |
> | PHC ResNet152 n=2    | 6.76 G | 3.38 G   |
> | PHC ResNet152 n=3    | 6.76 G | 3.38 G   |
> | PHC ResNet152 n=4    | 6.76 G | 3.38 G   |

---

> ### Author Response · Authors · 2021-11-22
> **On more investigations on activation functions and pruning**
>
> No, we did not consider investigating other activation functions or whitening procedures since our purpose is to compare our approach with a set of different real-valued baselines. We want to show that common neural models, such as ResNets or VGGs, can be endowed with our approach and obtain better results while reducing the number of parameters and storage memory required thanks to the hypercomplex algebra properties.
> Therefore, both for image classification and sound event detection tasks, it is crucial to have a fair comparison. We ensure this by leaving unchanged the hyperparameters and the architectural structures among the baselines and the models equipped with our method.
> As well, we only consider the standard batch normalization since whitening procedures should be specific to the domain and are still often approximations for the exact computation (https://arxiv.org/pdf/2104.09630.pdf). However, we added the references the Reviewer mentions in the paper revised version (Section 5).
> Finally, we did not consider matrix rings but it would be an interesting suggestion. However, our approach wants to build hypercomplex layers in a Hamilton-like form, which allows us to exploit the correlation between channels and thus saving a huge number of parameters.
>
>
> About pruning.
>
> We would like to thank the Reviewer for her/his suggestion. The proposed method defines a novel layer that can be employed in full or pruned networks. Indeed, our method aims at building novel modules that may replace every pre-existing convolutional layer, thus it is a more general approach with respect to pruning.
> However, pruning PHC models may be a very interesting idea that should be deeply investigated in a dedicated paper. Indeed, we may apply generic state-of-the-art pruning methods to our PHC models or thoroughly define tailored pruning approaches for PHC networks.
> Although, we added a dedicated section (see Appendix C of the revised paper) to include this interesting idea in the paper. Thanks for the suggestion.

---

### Official Review · Reviewer_6bRZ · 2021-11-03

**Correctness:** 2
**Technical Novelty And Significance:** 2
**Empirical Novelty And Significance:** 2
**Recommendation:** 5
**Confidence:** 2

**Main Review:**

Strengths:
The paper generalize the quaternion neural networks to CNNs and it shows better accuracy v.s. #parameters trade-off than Quaternion counterparts.

Weakness:
It's not clear how much real speedup/model size reduction can be achieved. The algorithms firstly use Kronecker product to generate the convolution kernel then use it for the normal conv2d. In my understanding, only the number of free parameters is reduced, instead of the practical resource consumption for inference / training.
The comparison is not enough to show the effectiveness of the proposed method. How is the proposed method compared to low-rank based conv kernel decomposition? How is it compared to the depth-wise & point-wise decomposition like in Mobilenets? Without the comparison to other commonly used DNN compression approaches, it's hard to understand the significance of the proposed method.

**Summary Of The Paper:**

This paper proposed a lightweight convolutional neural network based on the idea of quaternion neural networks. The paper extended the hypercomplex linear layers to convolution neural networks to apply to many multi-dimensional applications. The authors defined the parameterization of hypercomplex convolutional layers so that convolution neural networks can utilize the quaternion algebra to improve the parameter efficiency.

**Summary Of The Review:**

See Main Review.

---

> ### Author Response · Authors · 2021-11-22
> **Comparison of the proposed approach with respect to other decomposition and compression methods and discussion on the model size reduction**
>
> We would like to thank the Reviewer for her/his suggestions.
>
> Our method aims at parameterizing hypercomplex convolutional layers and building them in a Hamilton-like form. This allows us to leverage hypercomplex algebra properties that build the weight matrix by reusing parameters and sharing them among input dimensions. As a consequence, hypercomplex networks are able to capture correlations among dimensions of multidimensional inputs while, simultaneously, reducing the number of parameters.
> Thus, the proposed method is not merely a compression approach, but it is an efficient technique to parameterize hypercomplex layers and expand them to any nD domain of interest, thus grasping relations among input dimensions and consequently reducing the parameters and the storage memory required. Indeed, our layers are always faster than quaternion counterparts (Table 1 and Table 7), while obtaining better results. Therefore, our approach is more general and it can be applied to standard and large-scale methods but it could be also applied to compressed approaches. We have explained this better in the paper.
>
> The model size can be arbitrarily reduced to $1/n$. Theoretically, the only limit for reduction is that $n$ can be at most as large as the number of filters in the smaller layer. As an example, if the model has a stack of layers with filters [64, 256, 512, …], $n$ can be theoretically set up to 64 and thus reducing the parameters to $1/64$. However, we test up to n=16 with a saving of convolutional parameters and required storage memory of 94% (Table 3, and Table 12 in Appendix B.3.3), boosting inference time (Table 6). Such a reduction may downgrade the performance but it may be suitable for embedded applications that require fast inference and have strict constraints for memory. Thus, as metrics to evaluate the efficiency of our approach, we considered training and inference time, storage memory required for inference. Moreover, for completeness, we have added the FLOPs count in Table 2 and Table 7 of the revised version.
>
> Furthermore, our method should not be compared to approaches such as low-rank decomposition and depthwise or pointwise decomposition. Indeed, low-rank decomposition is solely aimed at dimensionality reduction while we propose to parameterize hypercomplex convolutions. Indeed, in our approach, we need a product decomposition that does not approximate the matrix but allows building it in a Hamilton-like form. The parameters reduction is a consequence of the Hamilton-like form of the matrix since one of the properties of the Hamilton product is to reduce the parameters by 75% by reusing them in the layer. Moreover, both depthwise and pointwise decompositions are based on the singular value decomposition, which is not suitable for our scope. The Kronecker product decomposition perfectly fits our case since it allows the implementation of the Hamilton product form.

---

> > ### Comment · Reviewer_6bRZ · 2021-11-30
> > **inference time of the proposed model**
> >
> > Thanks the authors for the response. From the FLOPs results in the response, I didn't see there is reduction by using the proposed method. Where does the inference speedup come from?

---

> > > ### Author Response · Authors · 2021-11-30
> > > **On the FLOPs count and inference time**
> > >
> > > We would like to thank the Reviewer for the possibility of clarifying this aspect. We take the opportunity to clarify and summarize the main differences.
> > >
> > > The FLOPs count is similar to one of the real-valued models since the dimension of the PHC weight matrix is the same as the weight matrix for real-valued models, so a PHC forward pass performs the same operations of a real-valued one, with an almost negligible increase due to the Kronecker products.
> > >
> > > Nevertheless, as shown in Fig.1, the PHC weight matrix is built by reusing smaller submatrices. Therefore, this mechanism allows our method to handle only 1/n parameters, saving inference time in loading/organizing weights. As an example, suppose to have a real-valued weight matrix $W \in R^{s \times d \times k \times k}$ and a PHC n=4 weight matrix $H \in R^{s \times d \times k \times k}$ with the same dimension, the matrix $W$ will have $sdk^2$ parameters while the PHC one will have $sdk^2/4$ thanks to the reuse of filter submatrices, each of dimension $\frac{s}{4} \times \frac{d}{4} \times k \times k$. This results in lower inference time despite the same number of operations. The filter reusing is underlined in Fig.1 and deeply investigated in Section 3.1 and Appendix A.2.
> > >
> > > Indeed, FLOPs computation is not a direct indicator for training and inference time since it does not take into account several other operations. An example of FLOPs misleading information is brought by quaternion models. Real-valued and quaternion-valued networks have the same number of FLOPs; however, quaternion models are far slower than the first ones since FLOPs do not include concatenation and reshaping operations that are involved in quaternion layers.

---

> > > > ### Comment · Reviewer_6bRZ · 2021-12-01
> > > > **How is the inference time measured?**
> > > >
> > > > Thanks the authors for explanation. My understanding is that the main benefits is saving the weights loading time before doing the convolution/matrix multiplication. Is this saved loading time for DISK-> RAM (CPU)? Or CPU -> GPU? I think typically people didn't count loading from disk as part of the inference time. In addition, if the weights tensor is always reconstructed before computation, the reconstruction time is actually a factor which could make inference time longer. Could you explain how the inference time is measured and maybe provide a breakdown time of this process? (e.g., weights loading time, weights reconstruct time, convolution time, etc).

---

> > > > > ### Author Response · Authors · 2021-12-01
> > > > > **Clarification on the inference time**
> > > > >
> > > > > We would like to thank the Reviewer for the valuable discussion.
> > > > >
> > > > > Regarding the details on the inference time, to be consistent with the literature, we consider inference time as the time required by the model to compute the output on the test set, as specified in Table 1 of the paper.
> > > > > So, we compute the inference time of the Python instruction:
> > > > >
> > > > > `outputs = model(inputs)`,
> > > > >
> > > > > as can be seen in the code released in the repository (anonymized for blind review process): https://anonymous.4open.science/r/HyperNets-CBBB/README.md.
> > > > > We double-checked the results on the paper with the Python module `time` (https://docs.python.org/3/library/time.html) and with the `tqdm` one (https://github.com/tqdm/tqdm).
> > > > > Both the modules confirm that in the largest part of the experiments our approach is faster than real-valued baselines despite the reconstruction of the weights tensor.
> > > > >
> > > > > Moreover, the Kronecker product decomposition has already been proved to be a fast inference method in previous works, such as  https://arxiv.org/pdf/2102.08597.pdf (see Table 3 and Section 5.2).
> > > > >
> > > > > Concerning the time for loading weights, we believe that fewer parameters may result in a lower time for loading weights for inference purposes. In particular, we believe that our method may be suitable for future embedded applications or devices.
> > > > >
> > > > > We would like to thank the Reviewer again for her/his inspiring and constructive comments that have led to an improvement of our work. Furthermore, the points discussed will certainly be a matter of investigation in the near future and will help us understand how and where to further improve our approach. We hope that the Reviewer appreciates our efforts and the results we have achieved.

---

### Author Response · Authors · 2021-11-23
**Summary of changes**

We would like to sincerely thank the Reviewers for their feedback and constructive recommendations that have allowed us to improve our work.
We report a brief summary of the main changes made in the revised paper according to the Reviewers’ comments.

* We performed additional large-scale experiments on the **ImageNet dataset**. We tested the real-valued baseline against its counterpart endowed with PHC layers, proving that our approach is able to gain comparable or better results while saving parameters. We added this experiment in the main body of the revised version.

* We computed **FLOPs count** for image classification and sound event detection models and we added the computations in the revised version of the paper. We show that with a small (rather insignificant) increase of FLOPs count our method can achieve improved performances while often requiring lower training and inference time.

* We computed additional experiments reducing the number of parameters of real-valued baselines by 75% to prove that a **lower number of parameters** does not lead to higher generalization capabilities. Indeed, the reduced versions of the models always obtain a lower accuracy with respect to their full counterparts. This proves that the improved scores of our models are due to the hypercomplex algebra properties and that they are not a consequence of parameters reduction. We added these additional results in the Appendix.

* We computed **training and inference time** for each model we considered. However, we decided to not add this information to the revised version of the paper in order to avoid redundancy.

* We added some **references** as related works to better clarify the background in which our method operates.

* In the main body of the revised version, we referred more clearly to the **implementation details** in the Appendix and to the online available code to allow a complete reproducibility of our results.

* We carefully improved the **writing quality** of the paper.

* We added an Appendix section on **future perspectives** of the proposed works, embracing some suggestions of the Reviewers on tentative future works.

* We better clarified the **novel contributions** of our work in the introduction section and in the main body of the revised paper.

* We better clarified the **maximum parameters reduction** that our approach allows and explained why it is a more general method with respect to other compression techniques. Indeed, the proposed approach can be involved in full as well as compressed models since its parameters reduction is due to hypercomplex algebra properties that share layer parameters capturing pre-existing correlations among input dimensions. As a consequence, our method saves parameters.

We have done our best to meet the valuable suggestions of the Reviewers, hoping to have addressed any concern and to have improved the overall quality of our work.

---

### Decision · Program_Chairs · 2022-01-20

**Decision:**

Reject

**Comment:**

In general, the reviewers appreciated the elegant concept behind the paper and the good results. However, they also raised considerable reservations about the significance of a method that decreases the parameter count but not necessarily computational efficiency (FLOPS) or memory. While the additional analysis that the authors provided definitely helps to understand the limitations of the method, the reviewers were in the end quite divided on the significance of the results. In addition, all reviewers agreed that the writing was in somewhat rough shape and needed improvement.

In summary, this is definitely a borderline paper, but given the current reviewer assessment, I would recommend that it is not quite ready for publication.